# Graffiti, Aging and Subcultural Memory—A Struggle for Recognition through Podcast Narratives

**Malcolm Jacobson**

Department of Sociology, Stockholm University, 10691 Stockholm, Sweden; malcolm.jacobson@sociology.su.se

**Abstract:** This article engages with the existential importance of subcultural memory for middle aged men. The social site is digital and consists of the first three Swedish graffiti podcasts where graffitied life courses are reflexively constructed through conversations. The empirical material gives unique insight into the construction of subcultural aging and self-identity and offers a critical reflection on theories of youth cultures. The results show that sharing memories of youth, crime and agency shapes the meaning of graffiti and subcultural cohesion. Retrospective narratives on personal development and increased reflexivity and self-control are constructed. Story telling has a long tradition in graffiti and social media has lately been incorporated within the subculture. As graffiti is a holistic practice, writers adopt many techniques to create graffiti personas, and podcasts, in addition to writing, have been established as a contemporary way to practice graffiti. The article illustrates how graffiti podcasting forms a mnemonic community where the meaning of graffiti is negotiated. Podcasts are memory sites in a struggle for individual and cultural recognition of what used to be labeled a deviant subculture.

**Keywords:** graffiti; podcasts; subcultural memory; aging; reflexivity; recognition

## 1. Introduction

> *"Well, nothing is more precious to anybody than the emotional record of his youth, and you will find the trail of my sleeve-worn heart in this completed play [ … ]. On the surface it was and still is the tale of a wild-spirited boy [ … ]. However, beneath that [ … ] it is a play about unanswered questions that haunt the hearts of people and the difference between continuing to ask them [ … ] and the acceptance of prescribed answers that are not answers at all, but expedient adaptions or surrender to a state of quandary."*
>
> Tennessee Williams [1], *The Past, the Present, and the Perhaps*

### 1.1. Mnemonic Community, Subcultural Memory and Graffiti Podcasts

How do you wish to be remembered? Who will remember you? What will you be remembered for? And how do you remember yourself? Graffiti writers use buildings and public transportation in a contest where fame and recognition are the rewards. With signatures and paintings, they inscribe themselves into the visual landscape including on advertising, public art and sanctioned messages such as road signs. Graffiti is often done without permission and during the last decades its ephemerality has been enhanced along with a growing graffiti removal industry.

However, while the marks of graffiti writers have been erased, their stories, myths and memories prevail. Graffiti writers adopt contemporary media technology to preserve and remember their work and to present narratives that challenge a discourse where their art is represented as dirt and their practice as destructive. In subcultural podcasts, middle aged graffiti writers engage in public conversations on the permanent imprint graffiti has made on their life courses. They recollect entering the path of graffiti as adolescents in the 1980s and 1990s, at the time identifying themselves with a youth culture. Now,

in their 40s and 50s they renegotiate the meaning of the subculture and reevaluate their memories and their adult identity. Podcast conversations disclose that memories of youth continue to have profound emotional importance throughout life. Recollections of shared experiences define the self-identity of participants in subcultures, like this excerpt from the Swedish podcast Other Side of the Track suggests:

> **Podcast host:** How would you say? When you saw graff, was it kind of a way for you to express yourself and be seen, like it was for so many others?

> **Klik:** Personally I believe that I was a nobody, I mean I was not like the coolest guy in class. I wasn't the best at anything, I had a hard time in school, or not hard time but I had dyslexia and didn't get any support. [ . . . ] [A]nd then I made a tag on a trashcan and the day after I met Flow and he was like: "who the fuck is that guy?". And you know you felt like, Damn! What would happen if I do three or four [tags]? So off course I got an identity through it, that I didn't have before.

Subcultural podcasts are public conversations that construct and re-negotiate biographies of graffitied life courses. The existential dimensions these talks have is reflected in the titles of podcasts such as "My Life in Letters". A podcast guest describes the construction of identity through graffiti as a matter of "life and death", a place where he "found love", and "a path towards life". Something that is significant for the graffiti writers looking back is that their continuing interest in graffiti goes against expectations, both their own and others. Hence, they feel a need to reflect on and discuss their memories of youth with one another. In doing this, their individual biographies are braided into a fabric of collective subcultural memory that maintains social cohesion over time.

Subcultural practice is often associated with youth. However, the aesthetic tastes and forms of expression developed in youth are not cut off when entering adult life. Subcultural experience is embodied and marks participants more or less for life. Throughout adult life, subcultural practice and style are maintained, transformed, and narrated. In another Swedish podcast, two middle aged graffiti writers remember how graffiti was presented to them when they were young as something kids do:

> [T]hat quote where some old New York writers said this thing: "graffiti is for kids", that it is done by kids, for kids, like childish. "If you are above 18", or 20, or what he said, then you should call it quits. However, that is kind of sad that is usually when you get good at it.

Their conversation revolves around the book Spraycan Art [2] which influenced graffiti writing adolescents around the world in the 1980s. These aging graffiti writers agree that graffiti is related to youth but are at the same time puzzled by the enduring grip it has on their adult identity. They point out a paradox inscribed in graffiti culture: graffiti is considered a youth culture, but takes years to master.

The fact that middle aged graffiti writers' reflections are now available in podcasts for the first time is due to two temporal progressions that converge: the first generations of graffiti writers have reached middle age, and the expansion of digital social media now encompasses oral communication.

### 1.2. Purpose, Research Question and Outline

This article explores the role of collective memory in social cohesion between men who have been part of the graffiti subculture for several decades. The purpose is to expand our understanding of how middle age masculine identities and social cohesion are constructed through mnemonic practices that recollect participation in a youth subculture. The research question posed is: *How do mnemonic practices reflexively construct aging and social cohesion in podcasts on graffiti?*

Theories on collective memory that contribute to understanding of aging and long-time commitment within subcultures are introduced in Section 2 below. After a brief history of graffiti, its relation to age, crime and gender is discussed. Section 2.2 covers the major topics and debates within subcultural research. Section 2.3 introduces memory studies, which views memory as a collective rather than

an individual entity. To investigate how a subculture defines its boundaries, this article employs the concept *mnemonic community* and analyzes how *mnemonic practices* construct joint subcultural memory. Social boundaries constitute overlapping mnemonic communities of different sizes, from subcultures to nations [3]. The concept *subcultural memory* is proposed for further analysis of how self-identity and social cohesion are constructed through reflexive narratives and biographies that are further discussed in Section 2.4. Section 3 describes the data, reflects on the absence of middle aged women in Swedish graffiti podcasts (at the time of the study), and discusses methodological implications of analyzing podcasts.

The first theme (Section 4.1) of the analysis investigates entrance to the mnemonic community and how middle aged male graffiti writers collectively remember seeing graffiti and becoming writers in their youth. In this they construct graffiti as something different in relation to expected behavior. The second theme (Section 4.2) looks closer at how subcultural difference and identity is performed and communicated through recollections of an unconventional life course. The third theme (Section 4.3) addresses how middle age masculine identity and conduct is constructed in contrast to memories of youth. Section 5 concludes by discussing how mnemonic practices within podcasts can be understood as a way to practice graffiti and in what ways graffiti podcasts are part of a struggle for recognition.

## 2. Theoretical Framework

### 2.1. Graffiti Art, Crime, Youth, Career, and Gender

In the late 1960s, an increased number of children and adolescents adopted pen names which they wrote on walls and public transportation in North American cities to receive fame and recognition [4–12]. *Subway graffiti* or *spraycan art* was seen as distinct from previous graffiti since its writers formed a social community with specific practices, language, aesthetics, and history [13]. In this article, "graffiti" or "subway graffiti" designates images and practices within this tradition. Individuals doing graffiti typically identify themselves as *writers*. A graffiti tag is both the written signature of the graffiti writer, their pen name, and the most common motif for the letters in their *master pieces*. Austin describes graffiti writing as a culture and a network with a rhizome-like structure without a singular root [4] (p. 38). Baldini states that street art (including graffiti) is to be defined "holistically" as a process of meaning making that may include practices additional to the actual creation of illegal images [14] (pp. 32–34).

When subway graffiti was introduced to a global audience in the early 1980s it was associated with an established discourse of graffiti as protest and revolt by the young against older generations [15]. There are plenty of examples of graffiti painted by adults with permission but to this day graffiti is most associated with youth and crime [8,12–16]. Participants in the graffiti community work on their careers and writing involves construction of self-identity [10,16]. Graffiti is practiced in different ways as writers grow older and as abilities learned through graffiti can be utilized outside the subculture [4,8,10,12,17]. Early studies on graffiti concluded that writers typically refrained from graffiti when approaching adult life [5,9,10]. Later studies observe that many adult writers continue their practice but conclude that they are more inclined to paint with permission [8,12]. Recently, this conclusion has been challenged; many adult graffiti writers continue to paint without permission [18].

As graffiti writers have grown older several studies have investigated the connection between age and gender in construction of masculine identity. Building on the well-established observation that most graffiti writers are male, Macdonald points out that several of the practices within graffiti such as bravery, competition, and crime have traditionally been associated with hegemonic masculinity [10]. However, Macdonald's conclusion that adult graffiti writers adopt a more mature adult identity has been qualified. Monto et al. instead find that graffiti can function as an outlet for "outlaw" masculinity for adult men who otherwise lead conventional and respectable lives [18].

## 2.2. The Subcultural

*Subculture* has a long history in the social sciences as a concept used to analyze and understand the cultural expressions of youth. Since the 1970s it is particularly associated with the Centre for Contemporary Cultural Studies (CCCS) [19,20]. The CCCS added class perspectives to theories that had analyzed subcultures as responses or solutions to structural and sociocultural strain [21–24]. The CCCS interpreted subcultural practices and aesthetics as symbolic resistance from working-class youths against the dominant culture and has been criticized for its overemphasis on structure [22]. Subcultural theory has also been challenged by the increased age of participants [25]. Youth culture is increasingly seen as cultures you attach to in youth and then have life-long relation to [26–28].

Post-subcultural theory discards the concept of subculture and instead employs the labels lifestyle, neo-tribes, and scenes [21,29,30]. These theories emphasize increased individual agency, fluidity, internal diversity, temporary commitment, and individual movement between groups [22,24]. Scholars who continue to utilize the concept of subculture instead argue that subcultures construct strong social cohesion through shared meaning and internal construction of difference towards the outside [10,16,23,24,31]. One way to conceptualize this is to understand subcultural practices and aesthetics as "sacred" among believers and thus contrasted with "profane" mainstream life [23,32]. Subcultural scholars tend to see subcultures not as fixed groups but as performances of subcultural boundary work and identity [23]. Hodkinson advocates using the concept subculture when investigating the "substance" of shared identities, meanings and practices and how subcultural participants themselves understand their distinctiveness [24] (p. 634). Hodkinson suggests that research on aging and "cultural journeys" can bridge the division between subcultural theory and post-subcultural theory.

## 2.3. Mnemonic Practices and Subcultural Memory

Introducing memory studies to subcultural analysis acknowledges that the social meaning and function of memories is anchored in present situated practice. Collective memory describes how the "active past" forms our identities [33] (p. 111). Memories of youth are active in adult life and aging is reflexively constructed in relation to aesthetic taste and cultural artefacts in which cultural memories are inscribed [26]. The concept *subcultural memory* presented in this article draws on Durkheim's [32] focus on the centrality of shared belief for social cohesion and Halbwach's [34,35] argument that memory is a collective act. Further *subcultural memory* draws on theories of self-identity and reflexivity according to which our actions are guided by interpretation of our biographies [36,37].

Memory is integral to human cognition and reflection as well as to culture [38,39]. Memory is fundamental for society since it concerns "how minds work together" [33] (p. 109). A focus on the present and fluid character of memory suggests that "mnemonic practices" in various social sites should constitute the case for sociological study rather than approaching collective memory as an object or a "mystical group mind" [33] (p. 112).

Through social relations, memory is both acquired and recalled in interaction with other people—face to face, mediated, and through material artefacts [34] (p. 38). Memories are "stored" in impersonal "sites" [3] (p. 291) and in intermediates such as different forms of texts, oral traditions, photography, material forms such as buildings, practices and individuals [34] (p. 222).

Among the social functions of collective memories are information diffusion, demonstration of competence, social cohesion, recognition, fame, and pleasure [40] (p. 174). For memory to have these functions, it needs to be narrated in a successful way [40] (p. 176), that is in a way that is understandable within a "mnemonic community" that we have been socialized into [3]. Participants in a mnemonic community share "social frameworks for memory" [34] (p. 28), they monitor "mnemonic others", and fight "mnemonic battles" [3] (p. 285–286,295). Memory is selective and influenced by power relations, constraints, and social norms [3,33,41–43]. Memories that confirm social bonds within a group are privileged [34] (p. 223). Social frameworks for memory are normative boundaries or "horizons" that define which memories are relevant for a community [3] (p. 286). Shared beliefs held by a community or subculture need constant social support [44] (p. 64). Senior members of a community

can function as "go-betweens" that connect generations mentally [3] (p. 291). New members of a group are incorporated into the community through the memories of the group's founding and history, and events previous to the shared understanding of the group's horizon tend to be neglected if not totally forgotten [3] (p. 287).

Mnemonic communities are generationally demarked [45] (pp. 296–297). Collective memory specifically addresses changes in identity over time such as the transition from child to adult [35] (p. 35). Biographies are narrated in a culture that demands us to construct difference and change in relation to different age classes such as "being young" and "being no-longer-young" [3,26,46]. Consequently, aging is not limited to the change our bodies undergo but it is very much a cultural construct, we are "being aged by culture" [46]. As we move through social space and time, we shift groups and reinterpret our biographies in accordance with the horizon of the group we currently interact with [44] (p. 56) [3] (p. 92). These "alternations" are existential challenges and involve the construction of self-identity [33] (p. 111) [34] (p. 38) [44] (p. 65).

Memory and identity are central for social construction of honor and shame [47,48]. Social rewards and resources as well as punishment are organized through processes of identification [37] (p. 168). Consequently, social identification concerns social status and pride and produces and reproduces hierarchy and stratification that is anchored in memory. Prior pain and shame that a group has experienced can be inherited and experienced by later members [3] (p. 290).

## 2.4. Reflexivity and Biographies

Although previous research on graffiti and other subcultures has focused on participants' accounts of activities through interviews and at times even particular narratives (e.g., Macdonald [10] and Ferrell [6]), there has been a tendency to treat these accounts at face-value rather than as narrated collective memories. Austin acknowledges that large parts of the graffiti history are built on anecdotes and fragments [4] (pp. 226–227). The writers who have been most prolific may dominate narration of subcultural memory while the vast majority of writers who pass through the graffiti community are forgotten.

The subcultural career or biography that is constructed around the alter ego of a writer's tag is anchored in narratives and memories of a writer's previous performances [16] (pp. 161–163). According to Giddens the name is a "primary element" in the biographies that construct self-identity [36] (p. 55). Like the writer Miss17 states: "To me, the graffiti is the story. The name tells the tale [...]. Graffiti is a game that anyone can play. Take a name, see what you can do with it." [49]. Implicit in this and similar statements is the fact that the tag as an alter ego constructs an additional identity or a "moral career" that may only partly overlap with the identity tied to the name the writer is referred to within other communities such as family, school, and work [4,5,10,16]. The tag can construct a parallel subcultural identity or a "superhero" persona that transgresses normative behavior and law [16]. In this the risk and dangers associated with graffiti as a crime are used as resources for construction of masculine identity [10,18].

Subjectivation as well as reflexivity consider how individuals are shaped and shape themselves and their conduct towards self and others in regard to power relations. According to cultural prescriptions demanding individual development, life is modelled into a certain form or an "aesthetics of existence" [16,46,50–52]. Through interpretation of memories and construction of biographies in relation to our present social milieu the self is held together as an identity over time [44] (p. 106). Struggles over memory influence a subject's ability for future action [42]. We need to have a sense of who we are, of our history, and of where we are going to be able to act [24,36,37,50–52]. However, biographies are not objective reports over a sequence of events that an individual has experienced. They are stories that can be told in many ways, they have a fictional character and are dramaturgical performances [16,36,53,54]. A life history perspective can unfold connections between individual biographies and social and cultural structures. Biographies offer insight to temporal and contextual meaning-making processes from the perspectives of the individuals under study [55] (p. 309).

## 3. Materials and Methods

Since 2015, the number of subcultural podcasts on graffiti has increased. Four podcasts in the United States and three in Sweden have produced over 500 h of conversation in four years. Sweden, together with the United States, has produced most graffiti podcasts. Sweden was one of several countries where the New York style of graffiti was established as a novel cultural expression some 35 years ago. Hence, the memories of Swedish graffiti writers reflect experiences of the global diffusion of subway graffiti.

This article analyzes podcast conversations with Swedish graffiti writers that started writing two or three decades ago. The sample includes writers that participated in the first decade of the Swedish graffiti subculture between 1984 and 1994. The aim is to help understand the meaning of aging within a subculture. The sample includes 29 episodes from the first three Swedish graffiti podcasts—Klotterpodden [Doodle Pod], Andra Sidan Spåret [The Other Side of the Track] (AST) and Svenska Graffare Podcast [Swedish Graff Podcast] (SGP). The episodes were produced between November 2016 and June 2019. Episodes are typically between 45 min and two hours long, and the participants that matched the sample criteria are approximately 35 to 50 years old (their actual age is not always disclosed). The analyzed podcasts are non-profit projects that are publicly available for free. They are recorded by amateur podcast producers within the Swedish graffiti community. The producers or hosts guide conversation as semi-structured interviews and decide which guests to invite, with typically one writer per episode. Introducing podcasts as data aims to complement previous research on subcultures that uses interviews and participant observation (see discussion in Section 2.2). This article methodologically treats subcultural podcasts as "natural" data since they are public conversations between graffiti writers where the researcher cannot steer the direction of conversation [56,57]. Hence, the power balance in problem formulation is shifted in favor of the participants. In contrast to interviews, podcasts offer many of the virtues Potter [58] associates with "natural talk"; they are rich in information and not steered toward the researcher's concepts. Because the author of this article has previously published books and magazines on graffiti that have influenced the discourse, the author chose not to do interviews for this study but rather to step back and remain a listener.

Treating podcasts as natural data does not imply that they offer direct representation of reality outside of podcasts. Rather the claim made is that podcast conversations are real social interaction, as Hall states: "Identities are [ . . . ] constituted within, not outside representation" [54] (p. 4). Consistent with Lynch, data is not assumed to present an unmediated reality [59] (pp. 533–535). However, unlike Lynch this article does not treat data as mere "reminders" of "naturally organized ordinary activities". Instead podcasts are treated as naturally organized activities in themselves. Listening to collective memories in podcasts allows the researcher to study a memory site where mnemonic practices construct self-identities and subcultural meaning as well as cohesion over time.

It is important to consider which graffiti writers are given voice in podcasts. Some of the hosts of graffiti podcasts explicitly state that one of their aims is to restore the memory of graffiti writers who have been forgotten or neglected and to contribute to their culture with recognition. All participants that matched the sample criteria are men, as inviting female writers to graffiti podcasts is rare. This reflects that graffiti has traditionally been dominated by men but also hints that females are not given voice [10,11].

Podcast episodes were transcribed and analyzed according to thematic analysis [60]. According to this methodology codes are tools closely connected to the "raw" data and themes are developed as an integrated part of the researcher's analysis and conclusions [61]. This was combined with narrative analysis to capture continuity as well as contradiction within statements. Narrative analysis investigates which roles people are ascribed and focuses on how people create meaning out of events rather than describing what happened [62]. Drawing on cultural sociology [63] this article investigates the meaning graffiti has for its participants. The article consequently engages in an interpretative analysis of how actors make sense of themselves in their cultural worlds [64,65].

## 4. Results

### 4.1. Entering the Mnemonic Community

This theme investigates how middle aged male graffiti writers remember becoming part of the graffiti community in their youth and considers the shared horizons within this mnemonic community. Podcast narratives typically depart from reconstructions of the biographies of the guests. These temporal narratives seek to arrange events in a linear sequence often starting with questions about the guest's first memory of graffiti. In the following quote from Svenska Graffare Podcast (SGP) the writer, Core, testifies about the impression a book with graffiti images made on him when he was an adolescent:

> I was really kind of a table tennis and wind surfing guy [laughs]. [ ... ] And then I was hanging out with [ ... ] this guy called Måns who had got a book from his parents, [ ... ] Danish Wildstyle [Graffiti]. And I was falling off the chair. You know [...] when I opened up the pages it was like boom [...]. When I saw it, it was like instant. I understood, this is what I will be doing. It just was a kind of mega-experience. [...] It was absolutely magical, damn how much we looked in that book. [...] It made a very strong impact on me and then we started sketching right away.

The graffiti writers participating in podcasts remember their first impressions of graffiti as intense and life changing moments. Writers recall being carried away by graffiti and often express that they lost control over their actions, Jeks recollects:

> That's interesting, how something just comes across so strong. You just have to do it [ ... ] It was really so evident, everywhere, it wasn't that graffiti came along, it was all around you, right, it was everywhere, once I got [ ... ] my eyes [ ... ] on it.

Writers remember encounters with graffiti in the 1980s as turning points that gave their life course a new direction. These turning points can be understood as processes of "cognitive transformations" of the self where unconscious influence is combined with conscious reflection and then intentional action [52] (p. 1000). Embarking on a graffitied life course is remembered both as a reflective decision and as due to unconscious influences through situated practice—an interplay between daily conduct and reflection over one's own biography that Giddens associates with human reflexivity [36] (p. 57).

These memories construct writers as participating in the early formation of European graffiti culture. As Zerubavel states, narratives of beginnings create shared meaning or "horizons" within a mnemonic community [3] (p. 287). Pioneers within a group receive status and thus narrate the shared past that creates social cohesion [3] (p. 291) [37] (p. 168).

The narrative exemplified by the quotes above can be summarized as: When I was young I was overwhelmed by a different aesthetic practice and I knew I could not resist entering its community. In response to the research question: when middle aged writers share similar memories in podcasts they construct collective memories resulting in social cohesion and reaffirmation of subcultural identity established in their youth.

Several writers recollect thinking similarly to Core about their future: "this is what I will do." The word "this" represents a "sacred" distinction where the subcultural is constructed as different from the "profane" mainstream [32]. "This" also signifies adopting a new identity as a graffiti writer in contrast to the identity before the epiphany of graffiti. The memory of following "this" different path will be further analyzed below as a collective representation that constructs subcultural identity and cohesion [23,54]. When Jeks recollects being ten years old in 1986 and seeing graffiti for the first time, it sounds like he is entering the gate to a cave of treasures, or like he has found a new sacred world:

> I'll never forget going down to the subway platform with my mother [ ... ], it had been open for a week [ ... ]. I remember it was my first impression of graff, I remember clearly that when you went down into that new station there was graffiti absolutely everywhere.

Middle aged writers recollect their younger self as standing at a crossroad. Their previous identity was associated with mainstream activities, such as table tennis. Such activities prior to experiencing graffiti would now be discarded or put outside of the "horizon" of what is relevant for the mnemonic community between graffiti writers [3] (pp. 286–287).

Jeks' statement illustrates a change of mnemonic community and consequently is boundary work that constructs group cohesion and self-identify. He was still a child accompanied by his mother when he saw the images that would direct his future life course. Change of group, such as when an adolescent leaves home, involves junctions between different mnemonic communities and these memories tend be particularly pregnant [34]. According to Mannheim, attaching to a new group entails a "fresh contact" and as such it demands re-evaluation of our mental "inventory" and renders a specific notice in our individual biographies [45] (p. 292–294).

Podcasts are memory sites within which narration of graffitied life courses is a mnemonic practice that constructs collective subcultural memory of shared experiences and temporal horizons. Intense recollections of graffiti as a life changing experience represent shifts of self-identity from the social community of family, school, or neighborhood to the subcultural community. Memories are narrated according to shared understandings that make them relevant to the group that shares them [34] (p. 38) [3] (p. 289). As adults looking back on their youth, graffiti writers reaffirm their identity with the subculture through recollections of a shared origin. Memories of youth are active fundaments for present middle age identities [26]. Biographies are retrospective reflections where events are presented as meaningful from the present horizon which also enables future action [3,24,36,54]. While much of the previous literature on subcultures has taken retrospective statements as facts, this article intends to bring attention to these memories narrated in a present situation [3,34,62]. Here memories of youth are approached as insights to construction of middle age identity rather than seen as facts about past youth identity.

*4.2. Following A Different Path*

This theme will look closer at how collective mnemonic practices construct narratives of graffitied life courses as paths through life that are different from the expected and ordinary, involving narratives on youth, progress, agency and crime. As described above, middle aged male graffiti writers remember the subcultural as something that gave their life course a certain direction (compare Campos [16] (p. 162), Giddens [36] (p. 5), Macdonald [10] (pp. 101–124)). In SGP the writer Brain reflects on how his life course and mind has been formed by graffiti:

> Yeah, I kind of chose a different path, [ … ] graffiti, the concept, the whole mentality. [ … ] So, my life is kind of defined by graffiti. It has followed me, it's kind of, it has always been around. [ … ] I've found myself, in what I'm doing and I'm so grateful for that every morning when I wake up. [I]'m still thinking in graffiti terms.

Recollecting a life course "defined by graffiti" constructs a consistent subcultural identity from youth to adult life. These memories construct self-identity and support an "aesthetics of existence" where one elaborates one's "own life as a personal work of art" and gives it "a certain form in which one could recognize oneself" and others [51] (p. 49).

Several writers describe growing up under vulnerable socio-economic conditions, graffiti is predominately associated with working class neighborhoods and recollections of poor school results are common. An exposed situation is often used as explanation for starting a career in graffiti writing, the narrative of the writer Moral is an example of this:

> If you go a little deeper into why … it was fun, to scribble and it was beautiful with graffiti, but the whole thing for me! Now I know it was a way to express, to find your expression, something you function with, because I did NOT function in school, like, in the expected way. [ … ] I could never sit still or be like silent. So school was like an impossible way to get in,

in some way. And club activities—football, hockey, that was … no! That wasn't possible. However, writing graffiti, which you did by yourself. [ … ] If there had been a coach, an adult trainer who had been out on [the subway station] and been like "yes now the trains come in guys, now you should do this … . It would not have been interesting at all. [ … ] Then it would have been: "eh this, eh what did he say, no I didn't hear what he said, but this seems boring, let's bounce".

In Moral's biography, distance to the adult world is constructed based on him not meeting the expectations of prescribed behavior. This interpretation is reflexively arrived at when constructing his biography some 35 years later [3,44,62]. The subcultural community among youths is emphasized by Moral as distinct since it allowed him to express agency and ability and gave him recognition:

Yes, and it is self-chosen, [ … ] that was the core of it for me and it actually worked for me [ … ]. I kind of found love within this. I mean community and warmth, along that path into the society, and that has since made up a foundation for my whole life. It made me feel that I was functioning and hence I could work [and] function with other stuff too.

**SGP:** Mm, you could perhaps have ended up in any subculture, where you find a context, a community and fellowship and all those bits or?

**Moral:** Exactly, [ … ] when you don't function in normal situations, you kind of have to find your own path. You cannot just sit in school and not function and get the worst results, and settle for it, then you have to find an alternative, an alternative path into life, really.

**SGP:** Then we are there again, about finding purpose in life. And several [podcast] guests have described that they were messy, chaotic, and creative as well. Then graffiti came as a perfect thing, that happened to fit you, and then they just went along.

Moral's recollection shows distrust towards the adult world. Examining his biography he argues that he needed to go his own way, in this case the path of the subcultural. This reoccurring narrative among middle aged graffiti writers can be summarized as: I was a youth astray that found recognition within the subcultural community through which I achieved a sense of direction and ability to move further through life. Through subcultural practice you can get to know yourself, your abilities, and your limitations [44] (p. 64). The subcultural is remembered as offering recognition and a remedy to the shame of not meeting expectations, of not fitting in, and of not being seen [47] (p. 152) [48] (p. 137). While arguing that the subcultural was something separate and distinct, the narratives about personal development paradoxically draw on broader normative cultural prescriptions to speak about difference and change over the life course [46].

As indicated previously, memories in podcasts construct graffiti as a sacred world – a community of youths contrasted to the adult world, and several writers recollect a generational mistrust [4,15]. Writers who adopt this narrative tend to interpret subcultures as solutions to problems. As Moral argues, he needed to follow the subcultural track to eventually function within society. Perspectives on subcultures as solutions have been criticized for drawing on structuralism and not acknowledging subcultural autonomy [10,23,24,26,31]. However, these recollections are not objective representations of previous events, they are performances of "communicated difference" that have meaning and function within the presently situated mnemonic practice where life courses are examined [3,23]. Through graffiti podcasts, writers construct narratives where they have overcome obstacles, and hence present their life courses as successful and graffiti as precious. These writers utilize podcasts to narrate individual biographies and shared subcultural timelines. When many individual biographies are presented in a sequence of episodes through podcasts, they are braided into a fabric constructing the meaning of aging within the subculture. This shared subcultural memory results in social cohesion between the members of the mnemonic community.

Another aspect where writers' narratives associate subcultural difference with agency and transformation of the self is in recollections of explorative, dynamic, and transforming relation to the urban landscape:

> **Brain:** Something is reflected, it's us that shape societies, like, they're formed by people who express themselves. [ ... ] And that's kind of what I like about graffiti, and when I listen to the other guys you [the SGP host] have interviewed, and I realize that they're out there painting trains and that, it's somehow as if they try to prove some kind of reality out there. It's as if it's not only about just sitting at home dreaming or anything. It's actually about realizing yourself out there.

Following the subcultural path is remembered as different from conventional life - it is constructed as active and in motion, an engaged relation to their environment that molded their individuality and identity [5] (p. 76).

> **Ligisd:** [Graffiti is] a journey of discovery and it's how you make life richer by discovering new things. If you don't do that, you're just stuck in the same pace all the time and life is too precious for that.

Writers recollect graffitied life courses as active and thus different from mainstream life that is constructed as petrified (compare with Hannerz [23] on punks). According to cultural sociology and subcultural theory the difference between the subcultural inside and its outside is not objective but rather symbolically created [23] (p. 15) [54]. The aesthetics and practices of graffiti are constructed as sacred and different from ordinary profane uses of cities and landscapes.

> **Jeks:** You get out there and discover the city.

> **SGP:** Yeah, that's what graffiti's all about, now that you're 40, a backpack with spray cans, going into the woods looking for a wall, shit, it's like being a child again.

> **Jeks:** It's still there, the desire for discovery, it's always part of the desire to discover and explore [ ... ] It's just [like] when we were younger [ ... ] you learned to, like, work towards something, you had a drive because you have a vision.

The narrative of life-long uncovering of the urban landscape builds on memories of youth, "being a child again", and still having the "desire for discovery." Youth and exploration are remembered as essential to graffiti and these middle aged writers present themselves as still "being young" [46]. Consequently, through their memories they construct the social meaning of age and perform cultural aging [26,46].

In the narratives analyzed above, graffiti is remembered as something that offered direction, movement, progress and meaning to life. A common way writers use to associate subcultural difference with this movement is to argue that illegal activity is central to graffiti.

> **Brain:** [W]e were looking for the thrill, it was never legal to write in the subway, if it had been I don't think we would have done it. We really wanted to fuck up the bloody subway and make sure that everyone in Stockholm could see that [ ... ] here [we] come like. The more the better.

A mnemonic community share a "social framework for memory" [34] (p. 38). This framework is a "system of collective representation" within which the norms, authenticity, and boundaries of the subculture are constructed, reaffirmed and diffused through performances of difference towards an "undifferentiated mainstream" [23] (p. 102) [54] (p. 530). The claim that authentic graffiti should be done without permission (illegally) can be understood as a normative script within the mnemonic community. In the memories of graffitied life courses, one of the most distinct patterns scripting how difference between graffiti and mainstream society should be narrated, is that of painting without permission as well as other types of crimes in relation to graffiti.

**Moral:** Yeah, it has to blaze, or rattle and crackle. That's what graffiti is to me, not necessarily a nice image. That's graffiti too, but, you know, well . . . er, it comes right out and assaults you.

**SGP:** Exactly, a lot has been said about it, that graff has to be illegal for it to have a soul, even though styles on a legal wall can be nice too, but it's not the same, really.

**Moral:** Yeah, yeah, yeah, it was only then you'd get that "aaah" feeling in your stomach. [ . . . ] [C]rossing the line, that's what did it, that's what you got into, that's what made you think "wow this is so cool", I just have to do it.

Here, Moral and SGP remember subcultural difference as "crossing the line" of the law. Subcultural consistency and identity over several decades is constructed on youth memories from the 1980s and 1990s. In podcasts, writers manifest what Baldini calls subcultural "subversiveness" not through unsanctioned writing on walls in the present but through recollections of doing this in their youth [14] (p. 31). Through this, they construct middle age identity, drawing on a particular version of masculinity emphasizing risk and rebelliousness [18].

To be able to share memories where graffiti is "a riot" that "blazes, rattles, and crackles" is a way for middle aged male writers to dramaturgically perform identity with the subcultural practice and its subversiveness (compare Alexander [54], Baldini [14], Campos [16] (p. 162)). Hence, it is not necessary to actively paint without permission in the present to remain a part of the graffiti culture. Furthermore, legitimate activities that are "holistically" [14] part of the art world of graffiti and its "field of cultural production" [66] (p. 19) can be seen as ways to do graffiti in addition to illegal painting.

As reflected by the above excerpts, writers dramaturgically perform graffiti through narratives in accordance with the shared "social framework" of the subcultural mnemonic community [34] (p. 38) [54]. As analyzed above, progress, agency and crime are central ingredients in these performances. Individual memories of youth get their meaning as collective representations of difference towards an undifferentiated mainstream.

*4.3. Middle Age Masculine Identities*

The previous two themes analyzed how male graffiti writers collectively remember entering and following a different path through life. This theme further investigates the meaning of memories for middle age masculine identity and for norms about present and future conduct. When writers describe their present relationship with graffiti, they state that graffiti continuously occupies their mind (compare Macdonald [10] p. 68). However, in contrast to recollections of their young selves many middle aged writers claim that they less often realize their ambition through actual spray painting. In the excerpt below the host at SGP asks the writer News about the meaning of graffiti in his present life. This question is stated in relation to a long conversation about history and memories.

**SGP:** How much emotion and mental activity do you have concerning [graffiti]?

**News:** It still takes, well I wouldn't say it takes time from me but I look at it constantly and it occupies my mind even if I don't, like, practice it. [ . . . ] You know, there is big drive and an interest all the time, even if you don't go along painting yourself.

News is an example of how youth experiences of graffiti have life-long influence on self-identity. Graffiti writing is inscribed in the self as a form of subcultural habitus. However, for many the way graffiti is practiced changes with age. In contrast to how middle aged writers recollect their youth, they argue that they as adults can control their impulses to paint. A common statement is that graffiti threatens to take more time than what seems reasonable in relation to other obligations (compare Campos [16] p. 163). Several writers describe painting as something they would like to do, or envision doing in the future, but often other things are more important. Similar to News, Ligisd shares that he is intrigued by seeing paintings by others, but says that his duties as a father reduce his actual graffiti writing:

You kind of have to try find a balance, it's like I would like to paint every day [ . . . ]. When you see others have been out painting or bombing then you get stoked. [...] [Y]ou think, tonight I might [be able to paint], and then when you come home – well not really, [ . . . ] you have to prepare gruel [for the baby] and other shit. [ . . . ] If you're in a relationship then you are in a relationship, then you must honor your own choice.

Writers claim that with increased age, they have achieved the insight that life includes more than graffiti and they argue that graffiti is not as important as it was in their youth. Similar to the quotes above, Moral often has the impulse to paint but like many others he qualifies this motivation as regularly being limited to an idea. This is due to the reflection that graffiti does not fill the same function in his adult life as when he was young.

**Moral:** I always have a feeling [ . . . ]. That you see a wall and think "ouff! Here you could [paint]". [ . . . ] But then you don't bother and you don't really have time to. And no motivation.

**SGP:** Ah! That you love graffiti but it's not your whole worldview in that way?

**Moral:** No, no, no, and even if it should be your whole worldview, it wouldn't matter. If I would paint every day, it wouldn't fill the same function for me.

Graffiti writers' gaze on the city is formed by life courses with graffiti occupying their minds, they see urban structures as possible sites to achieve fame and recognition [4,10]. However, as middle aged adults, they construct self-identity where painting graffiti is not allowed to be that central. Still, these writers' extensive reflections and recollections about graffiti illustrate that graffiti remains very important, but it has, for several of them, shifted from the execution of tags and pieces in the city towards recollections in podcasts and elsewhere. Through their narratives these graffiti writers reflexively construct the social meaning of aging. Through reflection on their biographies they connect past experiences and future aspirations with present conduct (compare Giddens [36] p. 54). Like many other podcast guests Brain argues that reflexivity increases with age:

Well, you know I think this is quite interesting because like later in life you start to kind of look at the paths you followed [ . . . ] and that's what I find interesting with your podcast [ . . . ], because now you have, like, time to reminiscence. It is sensitive as hell, when you embark on this [ . . . ] To tell your story [ . . . ] and kind of make sense of, how in hell did I get by [ . . . ]. And sometimes, I don't have a clue.

Brain's statement discloses that how to narrate your biography is not a given – as Hall argues identities have a fictional character [54] (p. 4). In middle age, writers express an increased need to reflect on their life course. They understand themselves as individuals who have different obligations and needs than when they were young. Adult identities that are in charge of the self are constructed in contrast to memories of younger less organized selves. The host of SGP expands on this topic, reflecting on his own challenges of self-control throughout different stages in life, first as a youth, and then a few years past, as an adult:

**SGP:** When I was younger I don't think I reflected over whether [graffiti was a choice or a need]. You know, I had a break. And then in adulthood a guy at my work found out [that] I had been painting [ . . . ]. He just said, "now we'll get some cans and paint". Then I knew, that if I take this step now, [ . . . ] well I was like a junkie with a shot of heroin in my hand, I knew that if I start now then I will fall down into it, devote a lot of time to it and maybe get into trouble again [ . . . ]. Then I had that insight, and I was correct, I am manic. I sit here with my graffiti podcast, but it's fun. [ . . . ]. [N]ow when you are an adult you can be responsible for your [graffiti] writing in a different way.

In this statement the overwhelming force of graffiti, previously discussed, is remembered as consistent since youth. Writing graffiti is here presented as a threat to the sanity of the adult subject. However, the shape the obsession takes is different to the recollections from youth, it is narrated as more internal and reflexive. For SGP, graffiti is now less about writing on walls, instead it is executed through obsessive production of podcasts where memories are narrated. Similar to other statements graffiti is increasingly something that occupies the mind.

Moral further expands on why graffiti does not fill the same function in middle age life as in youth. He contrasts the stable mid-life identity he has with the fragile identity beginning to take form in youth:

> [T]hat hunger and devotion that is graffiti, for me, that I could only have then, when I was a child you know. And that is what I find to be interesting with graff. [ . . . ] [Now] I can be a nerd about style as well, the shape of the letters and forms. [ . . . ] Because of this I can go to a legal wall [ . . . ] a couple of times a year with like some old writer friends, but then it is not at all about "ouff, what if everyone sees this, wow, here I am, just wow".

Moral constructs his present identity in contrast to his youth struggles of becoming someone. His statement indicating that contemporary graffiti is a social community that is, to a great extent, built on collective memories of life courses shared with "some old friends" and aesthetic tastes to "be a nerd about". These aesthetic tastes can be seen as "generational demarcations" of a "particular cultural milieu" [26] (p. 258).

When middle aged male writers in podcasts express personal development towards more self-control, their discussion involves norms for future behavior and how to be role-models for younger writers and for their own children. Writers present two ways to exercise self-control, which they argue have developed with increased age. First, to show consideration for others, primarily to their loved ones as Ligisd previously related. Second, to show consideration towards one's own well-being through knowledge of, and care for, the self [50]. While authentic graffiti is constructed as part of the outlaw life of young writers, the suggested ethos for middle aged male writers is to follow the path of the golden mean—or 'moderation in all things'.

> **SGP:** [M]aybe moderation in all things is the best, when it comes to, like, medium dangerous and fun stuff like graff and booze and . . .

> **Moral:** If one shall make any conclusion out of it, yes, but at the same time I am of the opposite opinion, graff was about being so illegal, so un-moderated as possible. [ . . . ] But this is not advice I give to others [ . . . ]. To my kids on the other side of the door I say "moderation in all things is good, do some graffiti on a legal wall, that's balanced". However, on the other hand there is a need for writers that are all but balanced, if graffiti should have any value so to say, [ . . . ] there should be some riot in it.

Moral contrasts bombing and illegal graffiti writing with painting with permission. Many of the most emotionally charged memories from youth that writers share in subcultural podcasts are from the illegal painting that they describe as authentic graffiti. When middle aged writers recollect and interpret their life courses, it is important for them to present graffiti as a productive experience while simultaneously not question the subcultural difference achieved by graffiti's subversiveness. Consequently, the narratives in graffiti podcasts present middle age masculine identities as results of subcultural youth experience leading to personal development and increased self-control and reflection. Critique against the practice of graffiti and its illegal acts are not within the normative boundaries that define which memories are relevant for this community (compare Zerubavel [3] p. 286). Constraints and social norms privilege memories that confirm social bonds within a group and discard memories that can be disruptive [34] (p. 223) [43] (p. 95).

## 5. Discussion

This article builds on close listening to 29 podcast episodes where middle aged male graffiti writers recollect graffitied life courses. They examine their youth from a middle age horizon and reflexively interpret their memories while the episodes are recorded. Listening to theses podcasts reveals the existential and emotional importance memories of youth have throughout life. Podcasts are an example of how digital social media has been more or less seamlessly integrated into subcultures. As Baldini argues the definition of graffiti cannot be reduced solely to painting without permission, but also includes other aspects of the lives of graffiti writers [14] (p. 33).

The analysis adopted an interpretative approach associated with cultural sociology and drew on narrative analysis to understand how actors make sense of events rather than focusing on descriptions of what happened [62–65]. The research question *"How do mnemonic practices reflexively construct aging and social cohesion in podcasts on graffiti?"* was approached through an analysis divided into three themes.

The analysis disclosed that through podcasts graffiti writers construct shared beginnings and thus horizons within a mnemonic community. Within this community collective memories construct social cohesion based on difference towards the subcultural outside. This difference is based on recollections of graffiti as an active, youthful and subversive practice that at the same time is productive for individual development. Writers argue that they have achieved increased reflexivity and self-control through graffiti. The analysis found that the middle age self is narrated as in charge of its actions, in contrast to the young self that is remembered as absorbed by graffiti. With increased age and reflexivity the practice of graffiti shifted from active writing towards retrospective construction of shared timelines and individual biographies. The analysis found that critique against the illegal aspects of graffiti is avoided since it could disrupt the community. This can be understood with Halbwachs' claim that collective memory only retains memories that are of pedagogic character for the group [34] (p. 223). Since subcultural memory and graffitied biographies cover several decades they construct temporal continuity. Subcultural identity and cohesion is confirmed even if the ways graffiti is practiced may change with the age of its participants and technical development. Podcasts reconfirm subcultural unity and reconnect aging writers who experience being scattered by non-subcultural adult obligations.

Two implications of the analysis stand out as particularly significant. Subcultural memories through graffiti podcasts are a struggle for recognition and podcasts can be seen as a way to practice graffiti. These conclusions are based on seeing graffiti from a holistic perspective as a field of artistic production or an art world [14,66]. Seen from this perspective the practice of graffiti includes many activities additional to writing on walls and podcasts are an example of this.

Graffiti is a practice where identities are constructed through writing and painting the names or tags that graffiti writers have adopted [5,10]. Graffiti *writers* can be seen as producing two forms of *text*, one is the actual writing on the wall, the other text is the narratives and myths constructed around tags, often in the form of memories. These masculine identities are built on narratives that construct "moral careers" or "superhero" personas [10,11,16,18]. To share recollections through a podcast under the alias of a tag is thus a way to construct meaning and fame associated with that identity. Hence, it is a way to write graffiti. Podcasting is also a way to practice graffiti since subcultural practice engages in struggles over memory and symbolic meaning [42]. Graffiti writers struggle for recognition of the value of their subculture and their life courses. Graffiti podcasts present narratives where graffiti contributes to personal development, hence the stigma of graffiti as an illegal practice is confronted. Writers turn shame over failures within other aspects of life into subcultural pride and recognition.

Podcasts offer rich material on subcultural memory and are particularly suitable when investigating how subcultural subjects want themselves and their community to be remembered. The narratives of middle aged graffiti writers do not constitute neutral observations on how subcultural life courses are directed. Middle aged male writers present their memories in a positive perspective to receive recognition. Listeners may sometimes ponder: "Are these memories romanticized idealizations?" The answer is that these memories are like the lives they recollect – reflexive narratives and dramatic performances structured by social frameworks within a mnemonic community [3,34,36,54].

Through graffiti podcasts, aging subcultural practitioners (re)construct internal norms, regulations and identities. Podcasts offer insights to how participants navigate boundaries as they move through social life and time. In graffiti podcasts, writers reflexively take charge of the meaning of their life and practice. They give their life a certain form and meaning. Their memories are emotionally charged with existential quandaries as they struggle for recognition of the dignity of their self and their culture.

**Funding:** This research received no external funding.

**Acknowledgments:** I would like to thank the graffiti writers who produced and participated in podcasts for the opportunity to listen in on their conversations. I hope that my analysis has done their struggle justice and will benefit their continuous reflections. I am grateful for the valuable comments from the anonymous reviewers. This article would not have been realized without discussions with, and help from: Susan Hansen, Erik Hannerz, Jacob Kimvall, Anna Lund, Fredrik Liljeros, Daniel Dahl, David Redmalm, Lauren Dean and Andy Bennett—thank you very much!

**Conflicts of Interest:** The authors declare no conflict of interest.

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
