# Peer review of "Graffiti, Aging and Subcultural Memory—A Struggle for Recognition through Podcast Narratives"

_societies, doi:10.3390/soc10010001_

Round 1

Reviewer 1 Report

The paper aims at elucidating mnemonic practices in middle ages graffiti writers and their role(s) in processes of identity construction and reconstruction. The paper argues for the relevance of remembering youth in construing writers' present middle-aged sub-cultural identities while re-interpreting the significance of the past. In particular, the author focuses on podcasts. The paper also goes a step forward and  suggests that sharing memories is also a way of practicing graffiti. 

The paper addresses a topic that is of interdisciplinary interest -- a topic that has not been widely discussed or examined in the scholarly literature. The author is clearly familiar with relevant literature on a varieties of sub fields that are here relevant -- including studies of sub-cultural identities, aesthetics of the self, etc. In this sense, the paper is likely to attract readers from a wide spectrum of disciplines.

I do have though some suggestions for improving the paper, which I believe could greatly profit from some writing and reworking. 

(1) Introduction: The introduction is perhaps the least effective section. It is way too long and dense. It is divided into a series of subsections making the experience of going through the beginning confusing and tiring. I suggest the author to streamline the intro by reducing to 1-2 pages. Section 1.6 is what really counts. The intro should also present a summary of the sections. The discussion in sections 1.2 - 1-5 should be moved to a separate theoretical section (something like "Theoretical Framework"), where author explains the key notions. I suggest to focus on notions and concepts that are central to the discussion, while avoiding historic reharsals of standard views that do not play a central argumentative role. 

(2) Identity in writers vs Identity in other subcultural groups: While reading, I was wondering whether the author somewhat believes that processes of identity construction for middle-aged writers differ significantly from what happens with other sub-cultural groups (e.g., punks). I do have a sense that the complex negotiation and re-negotiation of personal identities -- which are compatible with growing responsibilities, family lives, etc. -- that characterizes writers cannot be found in other (at least some) sub-cultural groups. Here perhaps the author may find conceptual resources in Ricardo Campos' work on writers' identities -- in particular, in his "Graffiti writer as superhero." The separation between two alter-egos (as Campos argues) may account for the peculiarities that I believe the paper hints at.

(3) Podcasts as doing graffiti: I do like the idea that remembering is a way of practicing graffiti. However, the claim is largely under-argued and not clarified. There's a sense in which one can take the claim as merely metaphorical. But it seems that author holds a stronger position -- which I find more interesting. Perhaps author wants to claim something along the following lines: the practice of graffiti cannot be reduced to the activities strictly connected with the creation of tags, throw-ups, and pieces, but encompasses a wide range of actions, deeds, features, etc., that are connected with the practice and constitute its meaning as a whole. Whether this reconstruction is correct is irrelevant, but author needs to explain this interesting (as well as controversial) claim. Author refers to Baldini (2018), who seems to endorse a position that could help author conceptualize the thesis. Baldini emphasizes that the practice of what he calls street art is to be seen "holistically," where processes of meaning-making involve an interplay between a larger set of actions, features, etc that cannot be identified with just the creation of visual works.

(4) Illegality: There is a bit of tension in the paper's discussion of illegality. At times, the paper seems to claim that illegality is not essential to graffiti, whereas at other points it seems to claim that authentic graffiti is illegal. Baldini (2018) provides a discussion of this issue that is perhaps less clear cut than the author here claims: illegality is central to the practice of graffiti (insofar as it grounds its subversiveness), but it is not necessary nor sufficient.  Writers' words as reported in the paper appear consistent with that view.     

Author Response

Response to reviewere 1

Thank you for very constructive and helpful comments!

Comment and suggestion 1:

Introduction: The introduction is perhaps the least effective section. It is way too long and dense. It is divided into a series of subsections making the experience of going through the beginning confusing and tiring. I suggest the author to streamline the intro by reducing to 1-2 pages. Section 1.6 is what really counts. The intro should also present a summary of the sections. The discussion in sections 1.2 - 1-5 should be moved to a separate theoretical section (something like "Theoretical Framework"), where author explains the key notions. I suggest to focus on notions and concepts that are central to the discussion, while avoiding historic reharsals of standard views that do not play a central argumentative role. 

Response 1:

I have followed the recommendations on moving section 1.2-1.5 and has also cut and restructured these parts. Section 1.1 has been rewritten to address comments from both reviewers. The introduction now includes a summary of the article (section 1.2.)

The comment on reiteration of “standard views” has been considered but regarded in the light of reviewere 2 commenting that the introduction assumed “prior knowledge” that the reader might not have. Consequently a very brief introduction to the background of “subway graffiti“ in the NYC tradition is maintained in section 2.1 (previously section 1.2).

In the theoretic framework the perspectives on graffiti as a “holistic” practice (Baldini) is discussed and the argument on “podcast as doing graffiti” is clarified. This together with other edits in analysis and discussion aims to also address comment 3 below.

Comment and suggestion 2:

Identity in writers vs Identity in other subcultural groups: While reading, I was wondering whether the author somewhat believes that processes of identity construction for middle-aged writers differ significantly from what happens with other sub-cultural groups (e.g., punks). I do have a sense that the complex negotiation and re-negotiation of personal identities -- which are compatible with growing responsibilities, family lives, etc. -- that characterizes writers cannot be found in other (at least some) sub-cultural groups. Here perhaps the author may find conceptual resources in Ricardo Campos' work on writers' identities -- in particular, in his "Graffiti writer as superhero." The separation between two alter-egos (as Campos argues) may account for the peculiarities that I believe the paper hints at.

Answer 2:

Since this article do not compare empirical findings between different subcultures I cannot conclude what is general for subcultures and what is specific for graffiti, I would assume that the importance of memories is general but their content would be specific according to the shared meaning of each subculture.

I have included Campos in the literary review and find that it strengthens some of my arguments on writer identity. Thanks for the suggestion!

Comment and suggestion 3:

Podcasts as doing graffiti: I do like the idea that remembering is a way of practicing graffiti. However, the claim is largely under-argued and not clarified. There's a sense in which one can take the claim as merely metaphorical. But it seems that author holds a stronger position -- which I find more interesting. Perhaps author wants to claim something along the following lines: the practice of graffiti cannot be reduced to the activities strictly connected with the creation of tags, throw-ups, and pieces, but encompasses a wide range of actions, deeds, features, etc., that are connected with the practice and constitute its meaning as a whole. Whether this reconstruction is correct is irrelevant, but author needs to explain this interesting (as well as controversial) claim. Author refers to Baldini (2018), who seems to endorse a position that could help author conceptualize the thesis. Baldini emphasizes that the practice of what he calls street art is to be seen "holistically," where processes of meaning-making involve an interplay between a larger set of actions, features, etc that cannot be identified with just the creation of visual works.

Answer 3:

In the theoretical framework and in the analysis I have clarified the writer’s relation to illegality building on a more accurate representation of Baldini’s argument. See also answer to comment 1 above.

 Comment and suggestion 4:

(4) Illegality: There is a bit of tension in the paper's discussion of illegality. At times, the paper seems to claim that illegality is not essential to graffiti, whereas at other points it seems to claim that authentic graffiti is illegal. Baldini (2018) provides a discussion of this issue that is perhaps less clear cut than the author here claims: illegality is central to the practice of graffiti (insofar as it grounds its subversiveness), but it is not necessary nor sufficient.  Writers' words as reported in the paper appear consistent with that view.     

Answer 4:

In the theoretical framework and in the analysis I have clarified the argument building on a more accurate representation of Baldini’s perspectives.

Additional changes in this version:

The headline is changed.

The abstract is shorter.

Reviewer 2 Report

The idea and background of this paper promised to be fascinating,with access to ageing graffiti artists and discussion of meanings and memories. But the paper itself needs significant reorganisation and revision. The reasons are as follows: 

The Introduction was confusing and began abruptly as if I had prior knowledge of this very particular area. The Introduction needs to be rewritten, actually introducing the paper and the study, and the theoretical framework of the study. Further down the paper you say "I am interested to see how collective memories construct self-identities ..." - this would be a good sentence to have in your Introduction. 

The Materials & Methods section reads like an early draft. The language needs some attending to, eg page 8 "instead of me asking about memories I listening to podcasts allow me to ..."

Is this a multi-authored work? If so, why is "I" referred to? 

All men and no problematisation of that, or discussion.

No exact ages for participants, just between 35 and 50 which is vague data. 

Quite a lot of subcultural theory is based on the memories of participants so saying that little exists is incorrect. What theories are you using and developing?

How are these collective memories if each participant was interviewed/recorded individually? 

I don't know who they are (names, occupation, age, years active, etc.) nor how you accessed them, nor what you asked them. Where did the recordings take place, when? 

You are halfway through the paper before you use quotes from your participants which seems a shame when you have so much rich data. 

Using themes is a good idea. I am not sure that your use of memory is successful. You could have more successfully focused on the writers' reflections and recollections of getting older, and how they fit into that subcultural world. 

At the end of the paper I have no sense of what your participants do or who they are. There is almost too much here, there are enough points for at least one more paper, which would give you opportunity to properly discuss fewer points in a more fluent and deeper way. 

Author Response

Response to reviewere 2

Thank you for constructive and helpful comments.

Comment and suggestion 1:

The Introduction was confusing and began abruptly as if I had prior knowledge of this very particular area. The Introduction needs to be rewritten, actually introducing the paper and the study, and the theoretical framework of the study. Further down the paper you say "I am interested to see how collective memories construct self-identities ..." - this would be a good sentence to have in your Introduction. 

Answer 1:

I have moved what was previous section 1.6 so it now is section 1.2. This aims to introduce the theoretical framework and the study earlier. This section discusses the topic of collective memory and self-identity. This change is also consistent with suggestions from reviewere 1.

Further section 1.1 has been edited to address the comment on assuming “prior knowledge”, however I have avoided adding a longer background of graffiti since reviewere 1 advised to avoid “historic rehearsals of standard views”. I have kept the first initial questions in the introduction since they suggest that the specific practice of graffiti has parallels with more general existential aspects of human life.  After this I have clarified the connection between graffiti writing and memory and my study.

As responded to comment and suggestion 8 one quote has been moved from results to introduction.

Comment and suggestion 2:

The Materials & Methods section reads like an early draft. The language needs some attending to, eg page 8 "instead of me asking about memories I listening to podcasts allow me to ..."

Answer 2:

This section has been partly rewritten

Comment and suggestion 3:
Is this a multi-authored work? If so, why is "I" referred to? 

Answer 3:

No, it is a single author, in the methods section it is no longer referred to “I” to make the article more consistent.

Comment and suggestion 4:
All men and no problematisation of that, or discussion.

Answer 4:

Discussion of this in method section is now revised

Comment and suggestion 5:

No exact ages for participants, just between 35 and 50 which is vague data. 

Answer 5:

I found publically available podcasts valuable data for sociological analysis. However using this data has some limitations for my knowledge of the subjects since I did not record the conversations myself, their age and occupation is not always disclosed. However, the analysis does not intend to draw conclusion from their age or occupation. The aim is to analyze what long term involvement in the graffiti subculture mean for participants. Consequently the sample is defined by the statements from writers that they started with graffiti in Sweden between 1984 and 1994.

This is now clarified in the methods section.

Comment and suggestion 6:
Quite a lot of subcultural theory is based on the memories of participants so saying that little exists is incorrect. What theories are you using and developing?

Answer 6

I agree, in the new version of the theoretic section I have now included two sentences in section 2.4 that clarify how this article analyzes memories as narrative constructs in contrast to a lot of the previous subcultural research that do not engage in the literature on memory studies.

Comment and suggestion 7:
How are these collective memories if each participant was interviewed/recorded individually? 

I don't know who they are (names, occupation, age, years active, etc.) nor how you accessed them, nor what you asked them. Where did the recordings take place, when? 

Answer 7:

I have rewritten parts of the theoretic section on collective memory (2.3) and the analysis  to clarify that they are collective memory since they build on shared understanding within the subculture and that podcasts episodes combined construct the meaning of graffiti through memories. I have edited the analysis and discussion with the same intention.

In the methods section (3) I have now clarified the conditions of the recordings (see also comment and suggestion 5).

In the analysis I have included more detailed about some of the questions that podcasts host states.

Comment and suggestion 8:

You are halfway through the paper before you use quotes from your participants which seems a shame when you have so much rich data. 

Answer 8:

I agree, I have moved one quote from the results to the introduction to give a more substantial introduction to the material. The theory section is now shorter. However, I did find that the theoretic section needs to be presented before the analysis. Since the analysis draws on several different research traditions the theory section is quite substantial.

Comment and suggestion 9:
Using themes is a good idea. I am not sure that your use of memory is successful. You could have more successfully focused on the writers' reflections and recollections of getting older, and how they fit into that subcultural world. 

At the end of the paper I have no sense of what your participants do or who they are. There is almost too much here, there are enough points for at least one more paper, which would give you opportunity to properly discuss fewer points in a more fluent and deeper way. 

Answer 9:

The themes aim to establish some of the key topics in podcasts on middle age graffiti writers. A lot of the conversation is about what happened to them when they were young, I have had the ambition to connect this with reflections on who they are now as adults. I have edited the analysis to make this clear and thus make my use of memory more convincing.

It is true that we don’t get a full picture of the participants, this is to a great deal due to the methodology (see answer to comment 5 and 7), however I hope that the benefits of the used methodology will balance this.

I have edited the analysis with the ambition to make each theme more specific and deep and hence make the arguments more focused.

Additional changes in this version:

The headline is changed.

The abstract is shorter.

Reviewer 3 Report

Thank you for the possibility to read your manuscript. Unfortunately, in my opinion, the manuscript has potential but, also, some fragilities. 

Some comments/suggestions:

-           I don’t agree that the “The article suggests that sharing memories is one of several ways to practice graffiti, in addition to the actual writing of graffiti.” (Abstract);

-           “Podcasts as a way to do graffiti” in the title seems excessive and not justified;

-           the abstract could be clearer and with a following structure: objective, methods, conclusion, implications;

-           structurally, the manuscript sometimes takes on the "tone" of an essay, other times as a research article. I suggest its clarification:

-           the keywords do not clearly reflect the manuscript;

-           “Subculture is a concept used to understand and analyze youth cultures.” Not only;

-           I suggest that the manuscript be reduced in length and made more precise in the justification of the respective research question: “The research question posed to reach this is: How do mnemonic practices reflexively construct aging and subcultural cohesion in podcasts on graffiti?”;

-           sometimes it seems to me that there is a certain temptation to force conclusions such as..: “Recording, participating in, and listening to podcasts is a way to practice graffiti because the language of graffiti, its traditions, and its norms are discussed, confirmed and reproduced through podcasts. . . “.

Author Response

Thanks a lot for helpful remarks that has improved the manuscript, I have commented on the edits made below.

Remark 1-3:

Title and abstract is changed to give a better representation of the article. The claim that podcasts is a way to do graffiti has been clarified in the analysis and discussion. I find this argument to be central for the analysis and reviewer 1 has previously found it justified.

Remark 4:

I have considered the remark about the tone and made edits to make the style more coherent. The manuscript will undergo copy editing and I will point out this remark to the person making the edits.  Please point out if there are specific examples where the tone is a major problem.

Remark 5:                     

I have updated the keyword to better reflect the central themes of the manuscript.

Remark 6:

The phrase is changed to: “Subcultural practice is often associated with youth.”

Remark 7:

The length is reduced and the I have edited the analysis and discussion to correspond to the research question that has been slightly edited (now it says “social cohesion” instead of “subcultural cohesion”).

Remark 8:

I have edited the text to avoid temptation to force conclusions. I have also added two sentences at the end of the methods section that clarifies that the analysis is made with an interpretative approach drawing on cultural sociological theory.

Round 2

Reviewer 3 Report

I recognize the ability of the authors to recognize several of my opinions/suggestions and incorporate them into the text. I don't always claim to be right, but the qualitative study has definitely become much more rigorous and perceptible. It was with great interest that I read this version of the text. It's demosntrate that: "Through graffiti podcasts aging subcultural practitioners (re)constructs internal norms,  . . .as they struggle for recognition of the dignity of their self and their culture.".

Edit the text, for example: "(Bennett 2010: 258):".

Author Response

Dear reviewer,

thank you for your helpful comments.

The English language and spelling has now been edited.